# Comparison of Gonadotropin-Releasing Hormone versus Estrogen-Based Fixed-Time Artificial Insemination Protocols in Grazing *Bos taurus* Suckled Beef Cows

**DOI:** 10.3390/ani13172803

**Published:** 2023-09-04

**Authors:** Luis B. Ferré, Julian Jaeschke, Juliana Gatti, Gerardo Baladón, Ezequiel Bellocq, Gustavo Fernández, Ramiro Rearte, Michael E. Kjelland, Marcos G. Colazo, Jordan M. Thomas

**Affiliations:** 1National Institute of Agricultural Technology (INTA), Chacra Experimental Integrada Barrow (MDA-INTA), Tres Arroyos B7500, Buenos Aires, Argentina; 2Departamento Técnico de Biogénesis Bagó S.A., Garin B1619, Buenos Aires, Argentina; julian.jaeschke@biogenesisbago.com (J.J.);; 3Servicio Técnico de Biogénesis Bagó S.A., Garin B1619, Buenos Aires, Argentina; 4Private Veterinary Practice, Indio Rico B7501, Buenos Aires, Argentina; 5Instituto de Investigaciones en Reproducción Animal (INIRA), Facultad de Ciencias Veterinarias, Universidad Nacional de La Plata, La Plata B1900, Buenos Aires, Argentina; 6Mayville State University, Mayville, ND 58257, USA; 7Department of Agricultural, Food, and Nutritional Science, University of Alberta, Edmonton, AB T6G 2P5, Canada; colazo@ualberta.ca; 8Division of Animal Sciences, University of Missouri, Columbia, MO 65211, USA; thomasjor@missouri.edu

**Keywords:** estrus synchronization, intravaginal progesterone device, prostaglandin, presynchronization, cattle

## Abstract

**Simple Summary:**

The use of estrogens in food animals is banned in several countries, but fixed-timed artificial insemination protocols (without the necessity of estrus detection) for beef cattle in South America are primarily based on estrogen. This study determined the optimal non-estrogen-based protocol in grazing *Bos taurus* suckling beef cows maintained under typical pasture conditions in Argentina. A total of 697 cows were enrolled in two experiments and the gonadotrophin-releasing hormone-based fixed-timed artificial insemination protocols resulted in similar or greater fertility compared to estrogen-based protocols in *Bos taurus* suckled beef cows. The greatest fertility was attained with the 7 & 7 protocol, which includes gonadotrophin-releasing hormone, prostaglandin and an intravaginal progesterone device. This study shows that non-estrogen-based fixed-timed artificial insemination protocols can successfully synchronize *Bos taurus* suckled beef cows under grazing conditions. Hence, beef cattle producers in South America could use the 7 & 7 fixed-timed artificial insemination protocol in order to meet those requirements imposed by countries that do not allow the use of estrogen in food animals.

**Abstract:**

Fixed-timed artificial insemination (FTAI) protocols for beef cattle in South America are primarily based on estradiol esters and intravaginal progesterone-releasing devices (IVPD). The objective of this study was to determine the optimal gonadotropin-releasing hormone (GnRH)-based protocol as an alternative to the use of estrogen-based protocols in grazing *Bos taurus* suckling beef cows. All cows received an IVPD on the day of protocol initiation and prostaglandin F_2α_ (PG) plus equine chorionic gonadotropin (eCG) treatments at the time of IVPD removal. In Experiment 1, cows (n = 235) were randomly assigned to one of four treatments: (i) 7-day estradiol = 2 mg of estradiol benzoate (EB) at IVPD insertion on Day 9 and 1 mg of estradiol cypionate (ECP) at IVPD removal on Day 2; (ii) 7-day GnRH = 10 µg of GnRH at IVPD insertion on Day 10, IVPD removal on Day 3 and GnRH at FTAI; (iii) 7 & 7 estradiol = PG at IVPD insertion on Day 16, EB on Day 9 and ECP at IVPD removal on Day 2; (iv) 7 & 7 GnRH = PG at IVPD insertion on Day 17, GnRH on Day 10, IVPD removal on Day 3 and GnRH at FTAI. In Experiment 2, cows (n = 462) were randomly assigned to one of four treatments: (i) 6-day estradiol = EB at IVPD insertion on Day 9, IVPD removal on Day 3 and GnRH at FTAI; (ii) 7-day estradiol; (iii) 7-day GnRH; (iv) 7 & 7 GnRH. In Experiment 1, plasma progesterone concentrations and percentage of cows with a corpus luteum (CL) at IVPD removal, and pregnancy per AI (P/AI) were greater for cows subjected to GnRH-based protocols compared with cows subjected to estrogen-based protocols (*p* < 0.01). In Experiment 2, cows subjected to the 7 & 7 GnRH protocol had the greatest P/AI (*p* < 0.01). In summary, GnRH-based FTAI protocols resulted in similar or greater P/AI compared to estrogen-based FTAI protocols in grazing postpartum *Bos taurus* suckled beef cows. The greatest P/AI was attained with the 7 & 7 GnRH protocol.

## 1. Introduction

In the last two decades, artificial insemination (AI) has significantly increased in beef cattle in South America. In this regard, the availability of estrogen esters in the veterinarian market has played an important role in the adoption of fixed-timed artificial insemination (FTAI [1]. However, the European Union (EU) prohibited the use of estrogens in food animals, and several countries have decided to adhere to this initiative in order to continue exporting animal products to the EU [2]. Therefore, in the near future, beef cattle producers in South America may be limited to the use of only non-estrogen-based FTAI protocols to meet those requirements imposed by the EU.

The most commonly used estrogen-based protocol consists of the insertion of an intravaginal progesterone-releasing device (IVPD) on Day 0 and administration of 2.0 mg of estradiol benzoate (EB; to synchronize follicular wave emergence). Prostaglandin F2α (PG; to ensure luteolysis), equine chorionic gonadotropin (eCG), and a dose of 0.5 or 1.0 mg of estradiol cypionate (ECP; to induce and synchronize an LH surge) is administrated at the time of IVPD removal 7 or 8 d later with FTAI typically done 48 to 56 h after ECP treatment [3]. Nonetheless, a more recently developed 6 d estrogen-based protocol, named J-synch, which provides for a lengthened proestrus is gaining popularity among beef producers in South America [4].

Protocols based on gonadotropin-releasing hormone (GnRH) have been used extensively for FTAI in dairy and beef cattle in North America [3]. A commonly used FTAI protocol in dairy cattle that utilizes two GnRH treatments given 9 d apart and a PG treatment given 7 d after the first GnRH has been called Ovsynch [5]. The CO-Synch is a modification of the 7-day Ovsynch protocol in which the second GnRH treatment is given concurrently with FTAI [6]. The latter protocol is more commonly used in beef cattle because it requires animal handling only 3 times.

With regard to physiology, the concentration of circulating steroid hormones and the size of the dominant follicle are determining factors in the release of LH and ovulation in response to GnRH administration [7]. It has been reported that 49 to 66% of beef cows ovulated following GnRH administration at random stages of the estrous cycle [6,8] and follicular wave emergence was synchronous only when GnRH induced ovulation [9].

Presynchronization with an IVPD and PG prior to a GnRH-based protocol increased the proportion of cows with a responsive follicle at the time of initial GnRH administration [8,10] and resulted in a greater proportion of cows having a functional corpus luteum (CL) at the end of the protocol [8]. A presynchronization strategy that includes administration of PG coincident with an IVPD insertion 7 d prior to a conventional 7-day CO-Synch protocol is known as 7 & 7 Synch [10,11].

The overall objective of this study was to determine the optimal GnRH-based FTAI protocol as an alternative to estrogen-based FTAI protocols in multiparous postpartum suckled (*Bos taurus*) beef cows maintained under typical pasture conditions in Argentina. We hypothesized that GnRH-based FTAI protocols would result in comparable pregnancy per AI (P/AI) compared to that obtained with the most commonly used estrogen-based FTAI protocols.

## 2. Materials and Methods

All experimental procedures followed internal Good Animal Welfare Practices established by National Institute of Agricultural Technology (INTA) Animal Care and Use Committee. All cows under study were treated ethically according to the Ethical Committee of the National Institute of Agricultural Technology (INTA).

### 2.1. Experimental Design

Experiments 1 and 2 were conducted at two different farms located within 50 km of each other. All cows were originally from those farms so hence acclimatisation was done already. Cows enrolled in both experiments were under similar herd management conditions and during the same breeding season. Both farms follow a complete preventive disease control plan supervised by a licensed veterinarian. The nutrition management consisted of ad libitum offering of alfalfa and ryegrass (grazing system), mineral salts and unrestricted access to water. Cows enrolled in this study were randomly assigned to each FTAI protocol based on body weight, days postpartum, body condition score (BCS) and ovarian structures before the initiation of this study but managed as a single group. The BCS was assessed at initiation of the FTAI protocol and at pregnancy diagnosis, using a 5-point scale as described by Houghton et al. (1990) [12]. The difference between both BCS measures was also calculated (BCS difference, Appendix A).

#### 2.1.1. Experiment 1

Multiparous (4–6 years of age) postpartum Red Angus suckled cows, weighing 504 ± 30 kg, with an average BCS of 3.7 ± 0.4 and 69 ± 11 d postpartum were enrolled in this experiment. Treatments and activities performed during Experiment 1 are shown in Figure 1. Cows were randomly assigned to one of four treatments: (i) 7-day estradiol (n = 59), 2 mg i.m. of estradiol benzoate (EB; BIOESTROGEN^®^, Biogénesis Bagó, Garin, Argentina) + IVPD containing 1.0 g of progesterone (CRONIPRES, Biogénesis Bagó, Garín, Argentina) on Day 9 then 1 mg of estradiol cypionate (ECP; CRONI-CIP, Biogénesis Bagó, Garín, Argentina) with 150 µg i.m. of D-Cloprostenol (PG; ENZAPROST DC, Biogénesis Bagó, Garín, Argentina) and 300 IU i.m. of equine chorionic gonadotropin, (eCG; ECEGON^®^, Biogénesis Bagó, Garín, Argentina) at IVPD removal on Day 2; (ii) 7-day GnRH (n = 59), 10.5 µg i.m. of buserelin acetate (GnRH; GONAXAL^®^, Biogénesis Bagó, Garín, Argentina) + IVPD on Day 10 then PG and eCG at IVPD removal on Day 3 and GnRH on Day 0; (iii) 7 & 7 estradiol (n = 60), PG + IVPD insertion on Day 16 then EB on Day 9, followed by ECP, PG and eCG at IVPD removal on Day 2; (iv) 7 & 7 GnRH (n = 57), PG + IVPD on Day 17 then GnRH on Day 10, followed by PG and eCG at IVPD removal on Day 3 and GnRH on Day 0. Cows underwent FTAI on Day 0, 50–54 h after IVDP removal for 7-day estradiol and 7 & & estradiol protocols or 60–66 h after IVPD removal for 7-day GnRH and 7 & 7 GnRH protocols.

#### 2.1.2. Experiment 2

Multiparous postpartum Black Angus suckled cows (4–7 years of age), weighing 400 ± 75 kg, an average BCS of 2.8 ± 0.4 and 66 ± 9 d postpartum were enrolled in this experiment. Treatments and activities performed during Experiment 2 are shown in Figure 2. Cows were randomly assigned to one of four treatments: (i) 6-day estradiol (n = 116), 2.0 mg i.m. EB + IVPD on Day 9 then 150 µg i.m. of PG and 300 IU i.m. of eCG at IVPD removal on Day 3, followed by GnRH on Day 0; (ii) 7-day estradiol (n = 115), as described in Experiment 1; (iii) 7-day GnRH (n = 116), as described in Experiment 1; (iv) 7 & 7 GnRH (n = 115), as described in Experiment 1. Cows underwent FTAI on Day 0, 72 h after IVPD removal for 6-day estradiol protocol, 50–54 h after IVDP removal for 7-day estradiol protocol or 60–66 h after IVPD removal for 7-day GnRH and 7 & 7 GnRH protocols.

### 2.2. Ultrasonography

Transrectal ultrasonography (Honda HS-102V equipped with a 5.0 MHz linear transducer, Honda Electronics Co., LTD, Tokyo, Japan) was done by one technician. The ovaries of all cows (Experiments 1 and 2) were examined just before the initiation of this study (Day 17) to determine ovarian structures. Cows with a CL were considered to be cycling (CY), cows without a CL and a follicle ≥ 10 mm were classified as no-CL with large follicles (No-CL/LF), and cows without a CL and follicles < 10 mm were classified as no-CL with small follicles (No-CL/SF) [13]. In Experiment 1, an ultrasonographic examination was also conducted on the day of IVPD removal to measure the diameter of the largest ovarian follicle and determine the presence/absence of CLs.

### 2.3. Blood Sampling and Radioimmunoassay

In Experiment 1, blood samples were collected by jugular venipuncture on Day 2 from cows subjected to 7-day estradiol or 7 & 7 estradiol protocols, and on Day 3 for cows subjected to 7-day GnRH or 7 & 7 GnRH protocols. Blood samples were collected in 10-mL vacutainer tubes coated with EDTA, kept on icepacks for <1 h and centrifuged at 700× *g* (3000 rpm) for 20 min. Plasma was subsequently stored at −20 °C until analysis. A radioimmunoassay (RIA) kit was used (IM 1188, Beckman Coulter, Immunotech, Brea, CA, USA) for the determination of plasma progesterone concentration [14], with all samples assayed in duplicates. The sensitivity of the assay was 0.15 ng/mL, and the intra-assay coefficient of variation (CV) was below 8% for concentrations between 0.15 and 60 ng/mL. The inter-assay CV was 4%.

### 2.4. Estrus Expression

An oil-based weather-proof formulated tail paint (Celotest, Biotay S.A., Buenos Aires, Argentina) was applied to all cows (Experiments 1 and 2) on the day of IVPD removal as an aid to determine estrus expression [15]. A uniform paint strip (approximately 20 cm long and 5–6 cm wide) was made using the applicator brush by applying constant pressure over the paint bottle along the spine in the same direction of the hair growth of the tail. The degree of paint removal was evaluated at FTAI. Estrus was defined to have occurred when the paint removal was ≥ 50%.

### 2.5. Artificial Insemination and Natural Service

All inseminations in Experiments 1 and 2 were done by one experienced technician using frozen-thawed semen from one proven fertility bull. The straws were placed in a Cito-thaw water bath (CITO Products, Watertown, WI, USA) at 38 °C for 45 s and the AI guns placed in an AI gun warmer (AI Electric Gun Warmer™, EM Tools, Inc., Rusk, TX, USA) until AI. A total of 12 to 14 days after FTAI, cows were exposed to bulls (bull-to-cow ratio, 1:30) for approximately 85 days. All bulls had passed a standard breeding soundness evaluation conducted by the herd veterinarian.

### 2.6. Pregnancy Diagnosis

Pregnancy diagnosis was done by transrectal ultrasonography (Honda HS-102V equipped with a 5.0 MHz linear transducer, Honda Electronics Co., LTD, Tokyo, Japan) 35 to 45 d after FTAI. The presence of a viable embryo/fetus (positive heartbeat) was used as a determinant of pregnancy. The final pregnancy status of all cows was determined by rectal palpation 90 days after bull removal.

### 2.7. Statistical Analysis

Data were analyzed using the GLIMMIX procedure in SAS (version 9.4; SAS Institute Inc., Cary, NC, USA).

The effect of the FTAI protocol on P/AI was assessed by fitting two logistic regression models, in each experiment. The first model included the fixed effects of FTAI protocols (Experiment 1:7-day estradiol, 7-day GnRH, 7 & 7 estradiol and 7 & 7 GnRH; Experiment 2: 6-day estradiol, 7-day estradiol, 7-day GnRH and 7 & 7 GnRH), estrus expression (yes or no) and their interactions. The second model included the fixed effects of FTAI protocols, ovarian status (CY, No-CL/LF or No-CL/SF) and their interactions. In both models, body condition indicators (initial BCS and BCS difference) were included, one at a time, and remained in final models if Akaike Information Criteria (AIC) decreased. The least squared means (LSM) of P/AI for each FTAI protocol, and their slice by the interaction term were estimated and the Fisher protected t test was used to estimate statistical significance for difference among LSM. Additionally, custom hypothesis test using contrast were applied to estimate the effect of type of FTAI (GnRH- vs. estradiol-based) protocol on P/AI.

In Experiment 1, the effect of FTAI protocol on largest follicle diameter and progesterone concentration at IVPD removal were assessed by fitting lineal regression models and the presence of CL was explained by fitting logistic regression models that included FTAI protocol categories as main predictor. Body condition indicators (initial BCS and BCS difference) were included, one at the time, and remained in final models if Akaike Information Criteria (AIC) decreased.

Experiment 1 had a statistical power of 80% to detect an increment in P/AI of 24% (i.e., 55% vs. 31% with a 95% confidence) among FTAI protocols; meanwhile Experiment 2 had a power of 80% to detect an increment in P/AI of 17% (i.e., 55% vs. 38%with a 95% confidence) among FTAI protocols.

## 3. Results

### 3.1. Experiment 1

The results of Experiment 1 are reported in Table 1 and Table 2, and Figure 3.

The LSM of the largest follicle and progesterone plasma concentration and the percentage of cows with a CL at IVPD removal for each FTAI protocol are shown in Table 1. The diameter of the largest follicle on the day of IVPD removal did not differ among FTAI protocols (*p* = 0.36). The percentage of cows with a CL at the time of IVPD removal differed (*p* = 0.001) among FTAI protocols. The percentage of cows with a CL was greatest in the 7 & 7 GnRH protocol, intermediate in the 7-day GnRH and 7 & 7 estradiol, and lowest in the 7-day estradiol. Circulating plasma progesterone concentration on the day of IVPD removal was greater in cows subjected to GnRH-based protocols compared with that in those subjected to estrogen-based protocols, especially in the presynchronized group of cows (*p* = 0.006).

Pregnancy per AI did not differ between GnRH-based protocols (*p* = 0.3), but both GnRH-based protocols resulted in greater P/AI than the 7 & 7 estradiol protocol (*p* = 0.001). Moreover, presynchronization before the GnRH-based protocol increased P/AI numerically, while presynchronization before the estrogen-based protocol reduced P/AI significantly (Table 2). The overall estrus expression rate was 75.3% and cows displaying estrus around FTAI had greater P/AI than those that did not display estrus (Table 2). The interaction between FTAI protocol and estrus expression indicates that in cows that did not display estrus the GnRH-based protocols resulted in greater P/AI in comparison to estrogen-based protocols (*p* = 0.008).

The percentage of cows with a CL at initiation of the FTAI protocols was 64.3%. There was an interaction between FTAI protocol and ovarian status. The 7 & 7 estradiol protocol resulted in lower P/AI than the GnRH-based protocols in cyclic and No-CL/SF cows (Figure 3). Cows subjected to the 7-day estradiol protocol had intermediate P/AI, which was not significantly lower than the P/AI in cows subjected to GnRH-based protocols. However, in No-CL/LF cows, the P/AI did not differ among FTAI protocols. The overall pregnancy (FTAI plus natural breeding) was 93.4%.

### 3.2. Experiment 2

Results of Experiment 2 are reported in Table 3 and Figure 4. Cows subjected to the 7 & 7 GnRH protocol had greater P/AI compared to all other FTAI protocols (Table 3).

The overall estrus expression rate was 85.5%. Cows displaying estrus around FTAI had greater P/AI than those that did not display estrus (Table 3). However, there was not interaction between FTAI protocols and expression of estrus (*p* = 0.305).

The percentage of cyclic cows at initiation of the FTAI protocols was 50.2% and cyclic cows had greater P/AI than acyclic cows (Figure 4). There was an interaction between FTAI protocol and ovarian status. In cyclic and No-CL/LF cows, P/AI did not differ among FTAI protocols. However, in No-CL/SF cows, the 7 & 7 GnRH protocol resulted in the greatest P/AI and the 6-day estradiol in the lowest P/AI. The P/AI in cows subjected to the 7-day estradiol or 7-day GnRH was intermediate. The overall pregnancy (FTAI plus natural breeding) was 94.1%.

## 4. Discussion

Our hypothesis that GnRH-based FTAI protocols in *Bos taurus* suckled beef cows would result in comparable P/AI to that obtained with the most commonly used estrogen-based FTAI protocols was supported. Moreover, both GnRH-based FTAI protocols tested in the current study significantly increased the percentage of cows with at least one CL at the time of IVPD removal and enhanced plasma progesterone concentration before FTAI.

The overall synchronization rate of current FTAI protocols is around 70–90%, in terms of synchronous emergence of a new follicular wave, presence of a functional CL at the time of PG treatment and synchronous ovulation of a new dominant follicle [16]. In the current study, cows with a CL present at the time of IVPD removal and PG treatment had greater estrus expression rate prior to FTAI, similar to observations in previous studies [10,11,17] and estrus expression is known to improve pregnancy success in beef [18,19] and dairy cattle [20] subjected to FTAI protocols. In this regard, the GnRH-based FTAI protocols resulted in a greater proportion of animals expressing estrus than did estrogen-based protocols, 80.2% vs. 70.6% and 87.4% vs. 83.5% for Experiments 1 and 2, respectively. Expression of estrus in cattle subjected to FTAI protocols may indicate the effectiveness of hormonal treatments, but also the ovulation of a mature follicle capable of producing adequate amount of estradiol. The expression of estrus is initiated by increased concentrations of estradiol [21], which makes favorable modifications to the uterine environment such as increasing uterine contractions for sperm transport [22], altering uterine pH to improve lifespan of sperm [21], and increasing oviduct secretions to augment fertilization rates [23,24]. Several factors and mechanisms are involved in the relationship between estrus expression and fertility as the ovulation of a mature follicle also results in a larger CL with greater progesterone synthesis [25,26].

In the present study, the 7 & 7 GnRH protocol resulted in the greatest P/AI after FTAI in comparison with a 7-day estrogen-based protocol (Experiments 1 and 2) or a 6-day estrogen-based protocol (Experiment 2) or a standard 7-day GnRH-based protocol (Experiment 2). Therefore, the results further demonstrated the efficacy of this presynchronization strategy using an IVPD plus the administration of PG, which generates a physiologically mature follicle, and therefore reduce the variability in the response to initial GnRH [8,10,11]. The GnRH-based protocols for FTAI rely on the initial GnRH treatment to induce ovulation and the emergence of a new follicular wave allowing a more synchronous dominant follicle at the time of IVPD removal and/or PG administration [27]. If ovulation does not occur in response to the initial GnRH treatment [28], estrus occurs prematurely (before PG) due to the lack of dominant follicle turnover [29], or animals fail to express estrus because a small, less physiologically mature follicle is induced to ovulate by the final GnRH administration [30]. The addition of an IVPD in cows treated with GnRH-based protocols reduces the occurrence of premature estrus among cows that failed to ovulate in response to the initial GnRH; however, this also increases the chances of the ovulation of an aged follicle [31] thereby contributing to the suboptimal P/AI observed in animals that still display standing estrus around FTAI.

Notably, the presynchronization strategy did not improve P/AI in the estrogen-based FTAI protocol (i.e., 7 & 7 estradiol protocol). A lack of consistency to induce a synchronous emergence of a new follicular wave after the administration of EB in the presence of low concentrations of circulating progesterone has been reported [32]. Therefore, we suspect that ovulation of an aged oocyte might be the main factor contributing to the reduced P/AI observed in cows synchronized with the 7 & 7 estradiol protocol [33].

As mentioned earlier, our results support the indication that estrus expression affects P/AI in animals subjected to FTAI [34]. Additionally, among cows expressing estrus, P/AI was greatest following GnRH-based protocols compared with estrogen-based protocols. Moreover, the 7 & 7 GnRH protocol resulted in the greatest P/AI in comparison to 7-day GnRH and estrogen-based protocols. Cows that exhibit standing estrus prior to FTAI had a larger dominant follicle with elevated circulating estradiol concentrations [35]. In addition, ovulatory follicle size has been reported to influence P/AI in suckled beef cows [30]. In this regard, increased diameter of the dominant follicle influences likelihood of ovulation and P/AI following FTAI in postpartum anestrus beef cows [1]. Administration of eCG at the time of IVPD removal could increase ovulatory follicle size, compensate for reduced ovulatory follicle size among cows that fail to ovulate following initial GnRH, and/or conceivably rescue follicle functionality [8]. This positive eCG effect may not affect all cows across all FTAI protocols to the same magnitude. For example, it is possible that cows receiving presynchronization and/or cows that are cycling before treatment may not benefit from eCG treatment [8], whereas eCG may be highly beneficial among acyclic cows and/or cows with low BCS [36]. Potential interactions of presynchronization, pre-treatment ovarian status, and eCG treatment should be further evaluated in future studies.

Conversely, cows not expressing estrus around FTAI may have an immature follicle resulting in delayed ovulations [37]. To overcome this downfall, in cows not expressing estrus by FTAI, it has been suggested that AI be delayed [38], and GnRH administrated prior to AI (split-timed AI) [39] or concurrent with AI [40]. In the present study, the combination of eCG and GnRH at the time of AI plus the overall very good BCS (Experiment 1) may have contributed to enhanced P/AI achieved in GnRH-based protocols in cows that failed to express estrus. Increased P/AI was also reported in estrogen-based protocols by the administration of GnRH in animals that did not show estrus 12 h before AI [41] or at the time of AI [42].

Although, different GnRH-based protocols for FTAI have been tested in *Bos indicus*-influenced beef cattle [43,44,45], results have been inconsistent and P/AI usually bellow that achieved with estrogen-based protocols [46,47]. Therefore, our results cannot be extrapolated to *Bos indicus* and additional research on the use of non-estrogen FTAI protocols in *Bos indicus* cattle is warranted.

Ovarian cyclicity status at the beginning of a FTAI program in suckled postpartum beef cows could influence likelihood of estrus expression and the synchrony of follicular development, thereby impacting P/AI [48]. In the present study, a greater proportion of cows expressed estrus following GnRH-based protocols, and expression of estrus was most likely among cows that were classified as cycling prior to synchronization, regardless of the FTAI protocol strategy used. Ovulation synchronization protocols contribute to improved reproductive efficiency in cattle by improving conception rates and increasing the proportion of eligible cows for breeding (submission rates) [49]. In South America, up to 60% of beef cattle could be in anestrus at the initiation of the breeding season [48]. Synchronization programs that maintain elevated progesterone during the growth of the ovulatory follicle through the use of IVPD [50] may contribute to improve the reproductive efficiency in postpartum suckled beef cows (especially in anestrus conditions) [51]. Anestrus cows that ovulate a dominant follicle after the first GnRH treatment had larger dominant follicle at IVPD removal [52]. In the present data, P/AI was only observed to be numerically improved in Non-CY/SF cows subjected to the 7 & 7 GnRH protocol, indicating that the presynchronization strategy in GnRH-based protocols can effectively increase fertility regardless of the initial ovarian status condition [53]. In addition, our results would suggest that protocols allowing a long exposure to IVPD such as the 7 & 7 GnRH protocol, may induce the resumption of cyclicity in Non-CY/SF cows. It has been shown that treatment with progestogens increased LH secretion in anestrus cows [54,55], particularly the attainment of a GnRH/LH pulse frequency [56,57,58].

In summary, the P/AI achieved with GnRH-based FTAI protocols was comparable to that with estrogen-based FTAI protocols in grazing *Bos taurus* suckled beef cows. Moreover, the 7 & 7 GnRH FTAI protocol consistently resulted in greater P/AI. A downside to the 7 & 7 protocol is that it requires one additional handling of cattle compared with other short-term FTAI protocols that use either GnRH or estrogen.

## Figures and Tables

**Figure 1 animals-13-02803-f001:**
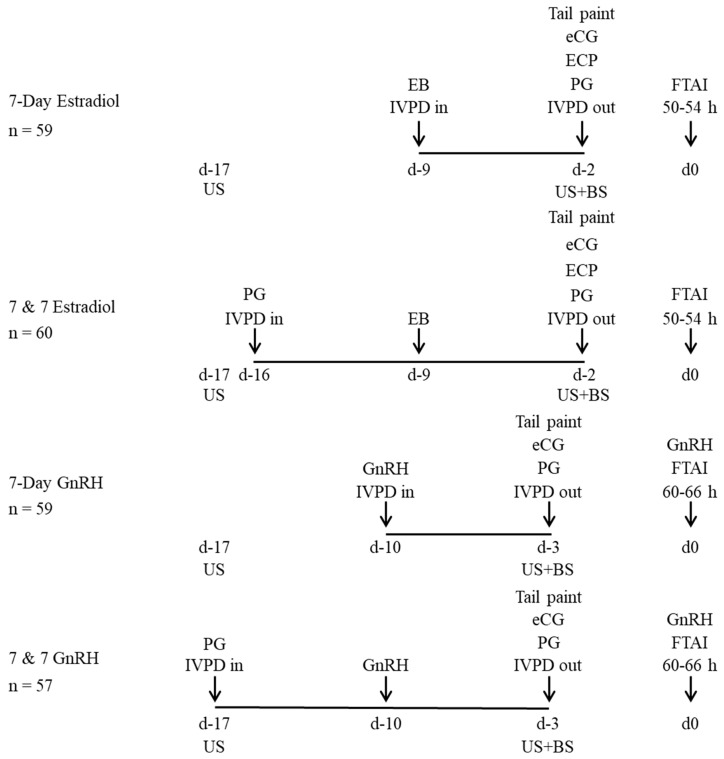
Schematic representation of treatments and activities during Experiment 1. Multiparous beef suckled cows were assigned randomly to one of four fixed-time artificial insemination (FTAI) protocols. IVPD: intravaginal device containing 1.0 g of progesterone, EB: estradiol benzoate (2 mg), eCG: equine chorionic gonadotropin (300 IU), ECP: estradiol cypionate (1 mg), PG: prostaglandin F2α (150 µg), GnRH: buserelin acetate (10.5 µg), US: B-mode ultrasound, and BS: blood sample.

**Figure 2 animals-13-02803-f002:**
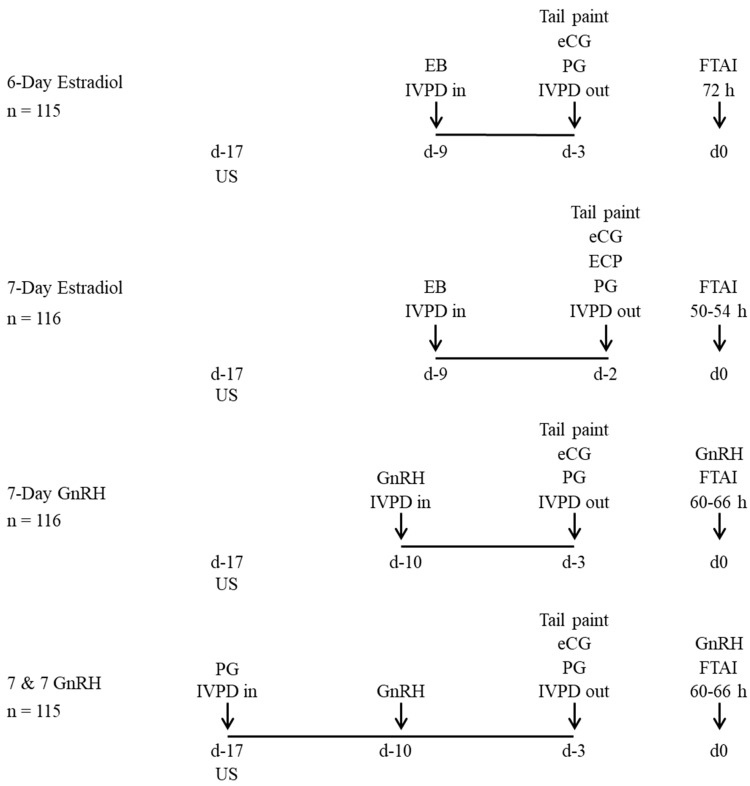
Schematic representation of treatments and activities during Experiment 2. Multiparous beef suckled cows were assigned randomly to one of four fixed-time artificial insemination (FTAI) protocols. IVPD: intravaginal device containing 1.0 g of progesterone, EB: estradiol benzoate (2 mg), eCG: equine chorionic gonadotropin (300 IU), ECP: estradiol cypionate (1 mg), PG: prostaglandin F2α (150 µg), GnRH: buserelin acetate (10.5 µg), US: B-mode ultrasound, BS: blood sample.

**Figure 3 animals-13-02803-f003:**
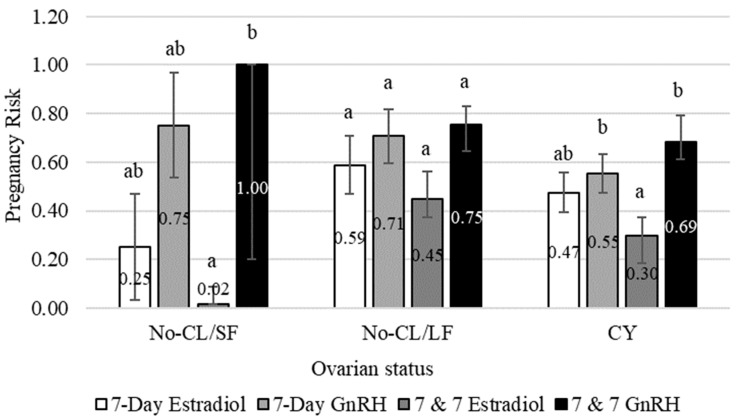
Least squares means for pregnancy risk estimated according to fixed-time artificial insemination (FTAI) protocol and ovarian status at initiation of the FTAI protocol (Experiment 1). The percentage of cows with a CL (CY) was 64.4, 64.4, 61.6 and 66.6% for 7-day estradiol, 7-day GnRH, 7 & 7 estradiol and 7 & 7 GnRH, respectively. The percentage of cows without a CL with a follicle ≥ 10 mm was (No-CL/LF) were 28.8, 28.8, 33.3 and 28.1% for 7-day estradiol, 7-day GnRH, 7 & 7 estradiol and 7 & 7 GnRH, respectively. The percentage of cows without a CL with follicles < 10 mm (No-CL/SF) was 6.7, 6.7, 5.0 and 5.2% for 7-day estradiol, 7-day GnRH, 7 & 7 estradiol and 7 & 7 GnRH, respectively. Pregnancy status was determined by transrectal ultrasonography 35 to 45 d after FTAI. ^a,b^ Bars with different superscripts within each cluster differ (*p* < 0.05).

**Figure 4 animals-13-02803-f004:**
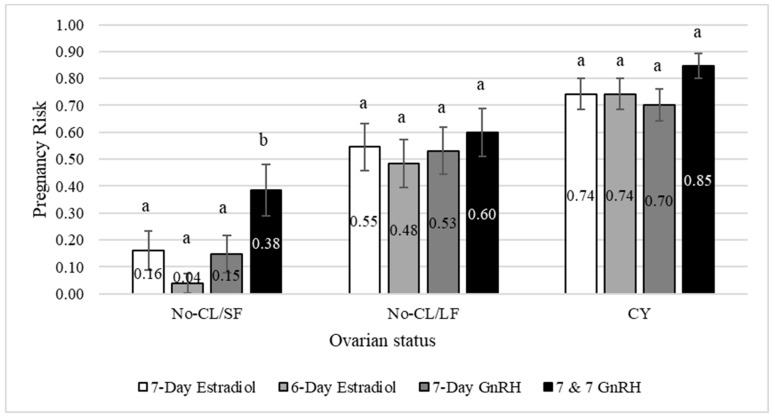
Least squares means for pregnancy risk estimated according to fixed-time artificial insemination (FTAI) protocol and ovarian status at initiation of the FTAI protocol (Experiment 2). The percentage of cows with a CL (CY) was 50.1, 50.4, 49.1 and 51.3% for 6-day estradiol, 7-day estradiol, 7-day GnRH and 7 & 7 GnRH, respectively. The percentage of cows without a CL with a follicle ≥ 10 mm (No-CL/LF) was 28.4, 26.9, 27.5 and 26.0% for 6-day estradiol, 7-day estradiol, 7-day GnRH and 7 & 7 GnRH, respectively. The percentage of and cows without a CL with follicles < 10 mm (No-CL/SF) was 21.5, 22.7, 23.3 and 22.6% for 6-day estradiol, 7-day estradiol, 7-day GnRH and 7 & 7 GnRH, respectively. Pregnancy status was determined by transrectal ultrasonography 35 to 45 d after FTAI. ^a,b^ Bars with different superscripts within each cluster differ (*p* < 0.05).

**Table 1 animals-13-02803-t001:** Least squares mean (LSM) and confidence interval (CI) of largest follicle, percentage of cows with a CL and plasma progesterone concentration at the time of intravaginal progesterone-releasing device (IVPD) removal according to fixed time artificial insemination (FTAI) protocol (Experiment 1).

	7-Day Estradiol	7-Day GnRH	7 & 7 Estradiol	7 & 7 GnRH
Total cows (n)	59	59	60	57
Largest follicle (mm) ^1^	11.8 ^a^	11.2 ^a^	11.0 ^a^	11.1 ^a^
[LSM (95% CI)]	(11.1–12.5)	(10.4–11.9)	(10.2–11.6)	(10.35–11.84)
CL presence (%) ^2^	57.6 ^a^	91.6 ^bc^	86.7 ^b^	98.2 ^c^
[LSM (95% CI)]	(46.3–71.8)	(84.5–99.1)	(78.4–95.8)	(94.5–100.0)
Progesterone (ng/mL) ^3^	8.3 ^a^	9.8 ^ab^	7.5 ^a^	11.3 ^b^
[LSM (95% CI)]	(6.4–10.3)	(7.9–11.7)	(5.6–9.4)	(9.3–13.2)

^a,b,c^ Within a row, values with different superscripts differ (*p* < 0.05). FTAI protocols: 7-day estradiol = cows were administrated 2 mg of estradiol benzoate (EB) + IVPD insert on Day 9 and 1 mg of estradiol cypionate (ECP) + IVPD removal on Day 2, 7-day GnRH = cows received 10 µg of GnRH + IVPD insert on Day 10, IVPD removal on Day 3 and GnRH at the time of FTAI, 7 & 7 estradiol = cows were administrated PG + IVPD insert on Day 16, EB on Day 9 and IVPD removal + ECP on Day 2, and 7 & 7 GnRH = cows received PG + IVPD insert on Day 17, GnRH on Day 10, IVPD removal on Day 3 and GnRH at the time of FTAI. ^1^ Largest follicle diameter (average of the height and width) measured by ultrasonography at the time of IVPD removal. ^2^ Presence of at least one CL determined by ultrasonography at the time of IVPD removal. ^3^ Plasma progesterone concentration at the time of IVPD removal.

**Table 2 animals-13-02803-t002:** Least squares mean (LSM) and confidence interval (CI) for estrus expression and pregnancy per AI (P/AI) according to fixed time artificial insemination (FTAI) protocol (Experiment 1).

FTAI Protocols		7-Day Estradiol	7-Day GnRH	7 & 7 Estradiol	7 & 7 GnRH
Total cows	n	59	59	60	57
P/AI ^1^	% (n)	49.1 (29) ^ab^	61.0 (36) ^bc^	33.3 (20) ^a^	71.9 (41) ^c^
	[LSM (95% CI)]	(37.8–63.8)	(49.7–74.9)	(23.3–47.8)	(61.1–84.7)
P/AI/Estrus expression ^2^	Yes (%)	61.5 ^b^	63.8 ^b^	40.0 ^a^	73.9 ^b^
	[LSM (95% CI)]	(48.0–79.0)	(51.4–79.3)	(27.9–57.3)	(62.2–87.8)
	No (%)	25.0 ^a^	50.0 ^ab^	13.3 ^a^	63.6 ^b^
	[LSM (95% CI)]	(11.7–53.6)	(28.3–88.3)	(03.6–48.8)	(40.6–99.7)

^a,b,c^ Within a row, values with different superscripts differ (*p* < 0.05). FTAI protocols: 7-day estradiol = cows were administrated 2 mg of estradiol benzoate (EB) + IVPD insert on Day 9 and 1 mg of estradiol cypionate (ECP) + IVPD removal on Day 2, 7-day GnRH = cows received 10 µg of GnRH + IVPD insert on Day 10, IVPD removal on Day 3 and GnRH at the time of FTAI, 7 & 7 estradiol = cows were administrated PG + IVPD insert on Day 16, EB on Day 9 and IVPD removal + ECP on Day 2, and 7 & 7 GnRH = cows received PG + IVPD insert on Day 17, GnRH on Day 10, IVPD removal on Day 3 and GnRH at the time of FTAI. ^1^ Pregnancy per AI (P/AI) determined by transrectal ultrasonography 35 to 45 d after FTAI. ^2^ Based on tail paint score at the time of FTAI. Yes ≥ 50% of paint rubbed off; and No ≤ 50 of paint rubbed off.

**Table 3 animals-13-02803-t003:** Least squares mean (LSM) and confidence interval (CI) for estrus expression and pregnancy per AI (P/AI) according to fixed time artificial insemination (FTAI) protocol (Experiment 2).

FTAI Protocols		6-Day Estradiol	7-Day Estradiol	7-Day GnRH	7 & 7 GnRH
Total cows	n	116	115	116	115
P/AI ^1^	% (n)	56.0 (65) ^a^	51.3 (59) ^a^	52.5 (61) ^a^	67.8 (78) ^b^
	[LSM (95% CI)]	(47.7–0.66)	(42.9–61.3)	(44.2–62.5)	(59.8–76.9)
P/AI/Estrus expression ^2^	Yes (%)	61.8 ^ab^	58.6 ^ab^	57.5 ^a^	71.0 ^b^
	[LSM (95% CI)]	(53.0–72.2)	(49.7–69.2)	(48.6–68.0)	(62.8–80.1)
	No (%)	25.8 ^a^	10.9 ^a^	23.6 ^a^	34.2 ^a^
	[LSM (95% CI)]	(12.1–55.2)	(03.0–40.5)	(10.0–55.6)	(15.4–76.0)

^a,b^ Within a row, values with different superscripts differ (*p* < 0.05). FTAI protocols: 6-day estradiol = cows were administrated EB + IVPD insert on Day 9, IVPD removal on Day 3 and GnRH at the time of FTAI. The 7-day estradiol, 7-day GnRH and 7 & 7 GnRH protocols as described for Experiment 1. ^1^ Pregnancy per AI (P/AI) determined by transrectal ultrasonography 35 to 45 d after FTAI. ^2^ Based on tail paint score at the time of FTAI. Yes ≥ 50% of paint rubbed off; and No ≤ 50 of paint rubbed off.

## Data Availability

The data that support the findings of this study are available from the corresponding author upon reasonable request.

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
