# Peer review of "Comparison of Gonadotropin-Releasing Hormone versus Estrogen-Based Fixed-Time Artificial Insemination Protocols in Grazing Bos taurus Suckled Beef Cows"

_animals, 2023, doi:10.3390/ani13172803_

Round 1

Author Response

Dear Dr. Augusto Madureira, Greetings!

Kindly, find the following comments after the revision of the manuscript entitled as, “Comparison of gonadotropin-releasing hormone- versus estrogen-based fixed-time artificial insemination protocols in grazing Bos taurus suckled beef cows”

Revision Note

Referee(s)' Comments to Author:

Referee: 1

Comments to the Author

The manuscript describes two experiments, using postpartum suckled beef cows, to determine the efficacy of estrous synchronization systems using GnRH to replace estrous synchronization systems which utilize estrogen compounds. This is important research as South American cow estrous synchronization systems traditionally have employed estrogen compounds which are effective but may limit opportunities to market products in the European Union.

We are thankful to the Reviewers for their constructive criticism to enhance the quality of the manuscript.

The comments have been addressed point by point in the revised manuscript.

We revised the manuscript using track changes function of word, so that, reviewers can identify the changes that we made in the revised manuscript. The Authors redoubled their efforts to answer all questions and clarify all doubts/inquiries.

Using a single point in time ultrasound examination to determine cyclic status is flawed. While the reviewer agrees that animals with a CL could be classified as cyclic, the authors are making large assumptions on cyclic status of animals that do not contain a CL. A better term for this observation is not cyclic status but ovarian status or ovarian structures at time of initiation of treatment. Ovarian status could be described as small follicles, large follicles and CL. Clarification is needed in the Materials and Methods on how “cyclicity” was used to allocate animals to treatment. It appears that cyclic status was not determined until treatments were actually being applied.

We appreciate this constructive feedback and have changed the wording as suggested. Authors replaced “ovarian cyclicity status” with ovarian structures or ovarian status accordingly. “Superficial anestrus” (cows without a CL and a follicle ≥ 10 mm) was replaced with “no-CL with large follicles (No-CL/LF)” and, “deep anestrus” (cows without a CL and follicles < 10 mm) was replaced by “no-CL with small follicles (No-CL/SF)”. All figures were replaced with the new terminology (including bars legends) in order to be consistent.

Were animals scanned then assigned to treatment based on ovarian characteristics in a rotating manner? How was equal distribution of different ovarian status categories to various treatments assured? Please clarify

L97 - How was cyclicity status determined ?

The authors acknowledge the lack of specificity and apologize for the unintentional error. All cows were ultrasound, weighed, recorded the days of postpartum and body condition score on day -17. All this info was classified in an Excel spreadsheet and all cows were randomly assigned (in a homogeneous and balanced way) to each of the FTAI groups. Authors replaced the schematic representation of treatments of both experiments to clarify the subject.

L51-57 - This is a run-on sentence which is hard for the reader to follow. It is confusing as to which injections are given at IVPD insertion and which are given at AVPD removal. Suggest splitting this into several sentences.

We have changed this as suggested.

L103 - indicate the scale for the BCS evaluation. It appears to be a 5-point scale. Other authors/countries may use a 9- or 10-point system. Please clarify in Materials and Methods.

Revised. In Lines 97-98 of the original manuscript (without track changes), authors specify the BCS scale used (1 to 5-point scale) and the reference.

L107 - suggest here and elsewhere in Materials and Methods to divide the description of procedures at IVPD insertions and those at IVPD removal by "with" or "then". For example, ".....1.0 g of progesterone on Day -9 with 1mg of estradiol cypionate ......."

Changed as suggested.

L333 - "increase" should be increases

Changed as suggested.

Table 2. – This is the only table or figure that combines the results from both experiments. While this reviewer can appreciate the desire of the authors to emphasize the benefit of GnRH-based systems in both experiments on P/AI, it does not follow the format of the other table/figure presentations. Furthermore, and more importantly, this table is unnecessarily complex and “busy”. Strongly suggest dividing this into 2 tables – one for each experiment.

Changed as suggested. Due to this modification, all tables have been modified as well.

While the authors are very clear that Bos taurus cattle are the focus of the experiments and subsequent application of results, a few additional sentences regarding the efficacy (or lack thereof) in Bos indicus cattle may enhance the Discussion. In addition, it would reinforce the need for additional research on non-estrogen containing synchronization systems for Bos indicus cattle.

Authors have now introduced a paragraph in the discussion referring to this point.

Reviewer 2 Report

The authors have done a nice job investigating the efficacy of options that are of high economic importance in areas of beef production that utilize estrogen in synchronization protocols. 

General Comments/Questions:

Ovarian activity categories were included in the model but no numbers for the distribution of animals in each category were reported in the actual text. While these numbers were presented in supplementary material, I believe it would increase the clarity of Figure 3 if some type of statement were given. 

In table 2 greater than 50% of animals no exhibiting estrus became pregnant for experiment 2 on the two GnRH protocols this is an interesting result but no discussion of this was included.

Specific/minor edits:

Lines 317 and 332 appear to have an extra space

Line 324 has references includes in 3 separate brackets.

Author Response

Dear Dr. Augusto Madureira, Greetings!

Kindly, find the following comments after the revision of the manuscript entitled as, “Comparison of gonadotropin-releasing hormone- versus estrogen-based fixed-time artificial insemination protocols in grazing Bos taurus suckled beef cows”

Revision Note

Referee(s)' Comments to Author:

Referee: 2

The authors have done a nice job investigating the efficacy of options that are of high economic importance in areas of beef production that utilize estrogen in synchronization protocols.

We are thankful to the Reviewers for their constructive criticism to enhance the quality of the manuscript.

Ovarian activity categories were included in the model but no numbers for the distribution of animals in each category were reported in the actual text. While these numbers were presented in supplementary material, I believe it would increase the clarity of Figure 3 if some type of statement were given.

Changed as suggested.

In table 2 greater than 50% of animals no exhibiting estrus became pregnant for experiment 2 on the two GnRH protocols this is an interesting result but no discussion of this was included.

Authors have now introduced a paragraph in the discussion referring to this point.

Lines 317 and 332 appear to have an extra space

Changed as suggested.

Line 324 has references includes in 3 separate brackets.

Changed as suggested.
